# Improving Textual Network Learning with Variational Homophilic Embeddings

**Wenlin Wang[1], Chenyang Tao[1], Zhe Gan[2], Guoyin Wang[1], Liqun Chen[1]**
**Xinyuan Zhang[1], Ruiyi Zhang[1], Qian Yang[1], Ricardo Henao[1], Lawrence Carin[1]**
[1]Duke University, [2]Microsoft Dynamics 365 AI Research
wenlin.wang@duke.edu

## Abstract

The performance of many network learning applications crucially hinges on the success of network embedding algorithms, which aim to encode rich network information into low-dimensional vertex-based vector representations. This paper considers a novel variational formulation of network embeddings, with special focus on textual networks. Different from most existing methods that optimize a discriminative objective, we introduce Variational Homophilic Embedding (VHE), a fully generative model that learns network embeddings by modeling the *semantic* (textual) information with a variational autoencoder, while accounting for the *structural* (topology) information through a novel *homophilic prior* design. Homophilic vertex embeddings encourage similar embedding vectors for related (connected) vertices. The proposed VHE promises better generalization for downstream tasks, robustness to incomplete observations, and the ability to generalize to unseen vertices. Extensive experiments on real-world networks, for multiple tasks, demonstrate that the proposed method consistently achieves superior performance relative to competing state-of-the-art approaches.

## 1 Introduction

Network learning is challenging since graph structures are not directly amenable to standard machine learning algorithms, which traditionally assume vector-valued inputs [4, 15]. Network embedding techniques solve this issue by mapping a network into vertex-based low-dimensional vector representations, which can then be readily used for various downstream network analysis tasks [10]. Due to its effectiveness and efficiency in representing large-scale networks, network embeddings have become an important tool in understanding network behavior and making predictions [24], thus attracting considerable research attention in recent years [31, 37, 16, 42, 8, 47, 40].

Existing network embedding models can be roughly grouped into two categories. The first consists of models that only leverage the *structural* information (topology) of a network, *e.g.,* available edges (links) across vertices. Prominent examples include classic deterministic graph factorizations [6, 1], probabilistically formulated LINE [37], and diffusion-based models such as DeepWalk [31] and Node2Vec [16]. While widely applicable, these models are often vulnerable to violations to their underlying assumptions, such as dense connections, and noise-free and complete (non-missing) observations [30]. They also ignore rich side information commonly associated with vertices, provided naturally in many real-world networks, *e.g.*, labels, texts, images, *etc*. For example, in social networks users have profiles, and in citation networks articles have text content (*e.g.*, abstracts). Models from the second category exploit these additional attributes to improve both informativeness and robustness of network embeddings [49, 36]. More recently, models such as CANE [40] and WANE [34] advocate the use of contextualized network embeddings to increase representation capacity, further enhancing performance in downstream tasks.

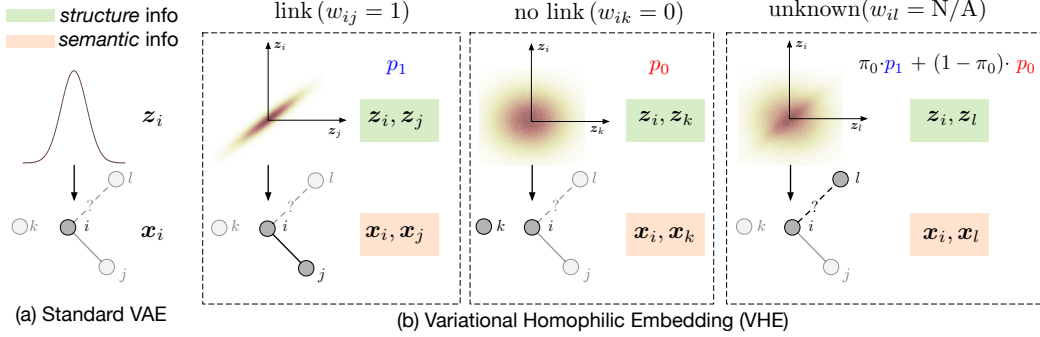

Figure 1: Comparison of the generative processes between the standard VAE and the proposed VHE. (a) The standard VAE models a single vertex $\boldsymbol{x}_i$ in terms of latent $\boldsymbol{z}_i$. (b) VHE models pairs of vertices, by categorizing their connections into: ($i$) link, ($ii$) no link, and ($iii$) unknown. $p_1$ is the (latent) prior for pairs of linked vertices, $p_0$ is the prior for those without link and $w_{ij}$ indicates whether an edge between node $i$ and node $j$ is present. When $w_{ij} = $ N/A, it will be sampled from a Bernoulli distribution parameterized by $\pi_0$.

Existing solutions, however, almost exclusively focus on the use of discriminative objectives. Specifically, models are trained to maximize the accuracy in predicting the network topology, *i.e.,* edges. Despite their empirical success, this practice biases embeddings toward link-prediction accuracy, potentially compromising performance for other downstream tasks. Alternatively, generative models [19], which aim to recover the data-generating mechanism and thereby characterize the latent structure of the data, could potentially yield better embeddings [13]. This avenue still remains largely unexplored in the context of network representation learning [23]. Among various generative modeling techniques, the *variational autoencoder* (VAE) [21], which is formulated under a Bayesian paradigm and optimizes a lower bound of the data likelihood, has been established as one of the most popular solutions due to its flexibility, generality and strong performance [7]. The integration of such variational objectives promises to improve the performance of network embeddings.

The standard VAE is formulated to model single data elements, *i.e.,* vertices in a network, thus ignoring their connections (edges); see Figure 1(a). Within the setting of network embeddings, well-known principles underlying network formation [4] may be exploited. One such example is *homophily* [29], which describes the tendency that edges are more likely to form between vertices that share similarities in their accompanying attributes, *e.g.,* profile or text. This behavior has been widely validated in many real-world scenarios, prominently in social networks [29]. For networks with complex attributes, such as text, homophilic similarity can be characterized more appropriately in some latent (semantic) space, rather than in the original data space. The challenge of leveraging homophilic similarity for network embeddings largely remains uncharted, motivating our work that seeks to develop a novel form of VAE that encodes pairwise homophilic relations.

In order to incorporate homophily into our model design, we propose *Variational Homophilic Embedding* (VHE), a novel variational treatment for modeling networks in terms of vertex pairs rather than individual vertices; see Figure 1(b). While our approach is widely applicable to networks with general attributes, in this work, we focus the discussion on its applications to textual networks, which is both challenging and has practical significance. We highlight our contributions as follows: ($i$) A scalable variational formulation of network embeddings, accounting for both network topology and vertex attributes, together with model uncertainty estimation. ($ii$) A homophilic prior that leverages edge information to exploit pairwise similarities between vertices, facilitating the integration of structural and attribute (semantic) information. ($iii$) A phrase-to-word alignment scheme to model textual embeddings, efficiently capturing local semantic information across words in a phrase. Compared with existing state-of-the-art approaches, the proposed method allows for missing edges and generalizes to unseen vertices at test time. A comprehensive empirical evaluation reveals that our VHE consistently outperforms competing methods on real-world networks, spanning applications from link prediction to vertex classification.

## 2   Background

**Notation and concepts**   Let $\boldsymbol{G} = \{\boldsymbol{V}, \boldsymbol{E}, \boldsymbol{X}\}$ be a network with attributes, where $\boldsymbol{V} = \{v_i\}_{i=1}^N$ is the set of vertices, $\boldsymbol{E} \subseteq \boldsymbol{V} \times \boldsymbol{V}$ denotes the edges and $\boldsymbol{X} = \{\boldsymbol{x}_i\}_{i=1}^N$ represents the side information (attributes) associated with each vertex. We consider the case for which $\boldsymbol{X}$ are given in the form

of text sequences, *i.e.*, $\boldsymbol{x}_i = [x_i^1, ..., x_i^{L_i}]$, where each $x_i^\ell$ is a word (or token) from a pre-specified vocabulary. Without loss of generality, we assume the network is undirected, so that the edges $\boldsymbol{E}$ can be compactly represented by a symmetric (nonnegative) matrix $\boldsymbol{W} \in \{0, 1\}^{N \times N}$, where each element $w_{ij}$ represents the weight for the edge between vertices $v_i$ and $v_j$. Here $w_{ij} = 1$ indicates the presence of an edge between vertices $v_i$ and $v_j$.

**Variational Autoencoder (VAE)** In likelihood-based learning, one seeks to maximize the empirical expectation of the log-likelihood $\frac{1}{N} \sum_i \log p_\theta(\boldsymbol{x}_i)$ w.r.t. training examples $\{\boldsymbol{x}_i\}_{i=1}^N$, where $p_\theta(\boldsymbol{x})$ is the model likelihood parameterized by $\theta$. In many cases, especially when modeling complex data, latent-variable models of the form $p_\theta(\boldsymbol{x}, \boldsymbol{z}) = p_\theta(\boldsymbol{x}|\boldsymbol{z})p(\boldsymbol{z})$ are of interest, with $p(\boldsymbol{z})$ the *prior distribution* for latent code $\boldsymbol{z}$ and $p_\theta(\boldsymbol{x}|\boldsymbol{z})$ the *conditional likelihood*. Typically, the prior comes in the form of a simple distribution, such as (isotropic) Gaussian, while the complexity of data is captured by the conditional likelihood $p_\theta(\boldsymbol{x}|\boldsymbol{z})$. Since the marginal likelihood $p_\theta(\boldsymbol{x})$ rarely has a closed-form expression, the VAE seeks to maximize the following evidence lower bound (ELBO), which bounds the marginal log-likelihood from below

$$\log p_\theta(\boldsymbol{x}) \geq \mathcal{L}_{\theta,\phi}(\boldsymbol{x}) = \mathbb{E}_{q_\phi(\boldsymbol{z}|\boldsymbol{x})}[\log p_\theta(\boldsymbol{x}|\boldsymbol{z})] - \mathrm{KL}(q_\phi(\boldsymbol{z}|\boldsymbol{x})||p(\boldsymbol{z})), \tag{1}$$

where $q_\phi(\boldsymbol{z}|\boldsymbol{x})$ is a (tractable) approximation to the (intractable) posterior $p_\theta(\boldsymbol{z}|\boldsymbol{x})$. Note the first conditional likelihood term can be interpreted as the (negative) reconstruction error, while the second Kullback-Leibler (KL) divergence term can be viewed as a regularizer. Conceptually, the VAE *encodes* input data into a (low-dimensional) latent space and then *decodes* it back to reconstruct the input. Hereafter, we will use the terms encoder and approximate posterior $q_\phi(\boldsymbol{z}|\boldsymbol{x})$ interchangeably, and similarly for the decoder and conditional likelihood $p_\theta(\boldsymbol{x}|\boldsymbol{z})$.

# 3 Variational Homophilic Embedding

To efficiently encode both the topological ($\boldsymbol{E}$) and semantic ($\boldsymbol{X}$) information of network $\boldsymbol{G}$, we propose a novel variational framework that models the joint likelihood $p_\theta(\boldsymbol{x}_i, \boldsymbol{x}_j)$ for pairs of vertices $v_i$ and $v_j$ using a latent variable model, conditioned on their link profile, *i.e.*, the existence of edge, via $\boldsymbol{W}$. Our model construction is elaborated on below, with additional details provided in the Supplementary Material (SM).

**A naïve variational solution** To motivate our model, we first consider a simple variational approach and discuss its limitations. A popular strategy used in the network embedding literature [10] is to split the embedding vector into two *disjoint* components: ($i$) a structural embedding, which accounts for network topology; and ($ii$) a semantic embedding, which encodes vertex attributes. For the latter we can simply apply VAE to learn the semantic embeddings by treating vertex data $\{\boldsymbol{x}_i\}$ as independent entities and then obtain embeddings via approximate posterior $q_\phi(\boldsymbol{z}_i|\boldsymbol{x}_i)$, which is learned by optimizing the lower bound to $\log p_\theta(\boldsymbol{x}_i)$ in (1) for $\{\boldsymbol{x}_i\}_{i=1}^N$.

Such variationally learned semantic embeddings can be concatenated with structural embeddings derived from existing schemes (such as Node2Vec [16]) to compose the final vertex embedding. While this partly alleviates the issues we discussed above, a few caveats are readily noticed: ($i$) the structural embedding still relies on the use of discriminative objectives; ($ii$) the structural and semantic embeddings are not trained under a unified framework, but separately; and most importantly, ($iii$) the structural information is ignored in the construction of semantic embeddings. In the following, we develop a fully generative approach based on the VAE that addresses these limitations.

## 3.1 Formulation of VHE

**Homophilic prior** Inspired by the homophily phenomenon observed in real-world networks [29], we propose to model pairs of vertex attributes with an inductive prior [5], such that for connected vertices, their embeddings will be similar (correlated). Unlike the naïve VAE solution above, we now consider modeling paired instances as $p_\theta(\boldsymbol{x}_i, \boldsymbol{x}_j|w_{ij})$, conditioned on their link profile $w_{ij}$. In particular, we consider a model of the form

$$p_\theta(\boldsymbol{x}_i, \boldsymbol{x}_j|w_{ij}) = \int p_\theta(\boldsymbol{x}_i|\boldsymbol{z}_i)p_\theta(\boldsymbol{x}_j|\boldsymbol{z}_j)p(\boldsymbol{z}_i, \boldsymbol{z}_j|w_{ij})\mathrm{d}\boldsymbol{z}_i\mathrm{d}\boldsymbol{z}_j. \tag{2}$$

For simplicity, we treat the triplets $\{\boldsymbol{x}_i, \boldsymbol{x}_j, w_{ij}\}$ as independent observations. Note that $\boldsymbol{x}_i$ and $\boldsymbol{x}_j$ conform to the same latent space, as they share the same decoding distribution $p_\theta(\boldsymbol{x}|\boldsymbol{z})$. We wish to enforce the homophilic constraint, such that if vertices $v_i$ and $v_j$ are connected, similarities between

the latent representations of $\boldsymbol{x}_i$ and $\boldsymbol{x}_j$ should be expected. To this end, we consider a *homophilic prior* defined as follows

$$p(\boldsymbol{z}_i, \boldsymbol{z}_j | w_{ij}) = \begin{cases} p_1(\boldsymbol{z}_i, \boldsymbol{z}_j), & \text{if } w_{ij} = 1 \\ p_0(\boldsymbol{z}_i, \boldsymbol{z}_j), & \text{if } w_{ij} = 0 \end{cases}$$

where $p_1(\boldsymbol{z}_i, \boldsymbol{z}_j)$ and $p_0(\boldsymbol{z}_i, \boldsymbol{z}_j)$ denote the priors with and without an edge between the vertices, respectively. We want these priors to be intuitive and easy to compute with the ELBO, which leads to choice of the following forms

$$p_1(\boldsymbol{z}_i, \boldsymbol{z}_j) = \mathcal{N}\left( \begin{bmatrix} \boldsymbol{0}_d \\ \boldsymbol{0}_d \end{bmatrix}, \begin{bmatrix} \boldsymbol{I}_d & \lambda \boldsymbol{I}_d \\ \lambda \boldsymbol{I}_d & \boldsymbol{I}_d \end{bmatrix} \right), \quad p_0(\boldsymbol{z}_i, \boldsymbol{z}_j) = \mathcal{N}\left( \begin{bmatrix} \boldsymbol{0}_d \\ \boldsymbol{0}_d \end{bmatrix}, \begin{bmatrix} \boldsymbol{I}_d & \boldsymbol{0}_d \\ \boldsymbol{0}_d & \boldsymbol{I}_d \end{bmatrix} \right), \quad (3)$$

where $\mathcal{N}(\cdot, \cdot)$ is multivariate Gaussian, $\boldsymbol{0}_d$ denotes an all-zero vector or matrix depending on the context, $\boldsymbol{I}_d$ is the identity matrix, and $\lambda \in [0, 1)$ is a hyper-parameter controlling the strength of the expected similarity (in terms of correlation). Note that $p_0$ is a special case of $p_1$ when $\lambda = 0$, implying the absence of homophily, while $p_1$ accounts for the existence of homophily via $\lambda$, the *homophily factor*. In Section 3.3, we will describe how to obtain embeddings for single vertices while addressing the computational challenges of doing it on large networks where evaluating all pairwise components is prohibitive.

**Posterior approximation** Now we consider the choice of approximate posterior for the paired latent variables $\{\boldsymbol{z}_i, \boldsymbol{z}_j\}$. Note that with the homophilic prior $p_1(\boldsymbol{z}_i, \boldsymbol{z}_j)$, the use of an approximate posterior that does not account for the correlation between the latent codes is inappropriate. Therefore, we consider the following multivariate Gaussian to approximate the posterior

$$q_1(\boldsymbol{z}_i, \boldsymbol{z}_j | \boldsymbol{x}_i, \boldsymbol{x}_j) \sim \mathcal{N}\left( \begin{bmatrix} \boldsymbol{\mu}_i \\ \boldsymbol{\mu}_j \end{bmatrix}, \begin{bmatrix} \boldsymbol{\sigma}_i^2 & \boldsymbol{\gamma}_{ij} \boldsymbol{\sigma}_i \boldsymbol{\sigma}_j \\ \boldsymbol{\gamma}_{ij} \boldsymbol{\sigma}_i \boldsymbol{\sigma}_j & \boldsymbol{\sigma}_j^2 \end{bmatrix} \right), \quad q_0(\boldsymbol{z}_i, \boldsymbol{z}_j | \boldsymbol{x}_i, \boldsymbol{x}_j) \sim \mathcal{N}\left( \begin{bmatrix} \hat{\boldsymbol{\mu}}_i \\ \hat{\boldsymbol{\mu}}_j \end{bmatrix}, \begin{bmatrix} \hat{\boldsymbol{\sigma}}_i^2 & \boldsymbol{0}_d \\ \boldsymbol{0}_d & \hat{\boldsymbol{\sigma}}_j^2 \end{bmatrix} \right), \quad (4)$$

where $\boldsymbol{\mu}_i, \boldsymbol{\mu}_j, \hat{\boldsymbol{\mu}}_i, \hat{\boldsymbol{\mu}}_j \in \mathbb{R}^d$ and $\boldsymbol{\sigma}_i, \boldsymbol{\sigma}_j, \hat{\boldsymbol{\sigma}}_i, \hat{\boldsymbol{\sigma}}_j \in \mathbb{R}^{d \times d}$ are posterior means and (diagonal) covariances, respectively, and $\boldsymbol{\gamma}_{ij} \in \mathbb{R}^{d \times d}$, also diagonal, is the *a posteriori* homophily factor. Elements of $\boldsymbol{\gamma}_{ij}$ are assumed in $[0, 1]$ to ensure the validity of the covariance matrix. Note that all these variables, denoted collectively in the following as $\phi$, are neural-network-based functions of the paired data triplet $\{\boldsymbol{x}_i, \boldsymbol{x}_j, w_{ij}\}$. We omit their dependence on inputs for notational clarity.

For simplicity, below we take $q_1(\boldsymbol{z}_i, \boldsymbol{z}_j | \boldsymbol{x}_i, \boldsymbol{x}_j)$ as an example to illustrate the inference, and $q_0(\boldsymbol{z}_i, \boldsymbol{z}_j | \boldsymbol{x}_i, \boldsymbol{x}_j)$ is derived similarly. To compute the variational bound, we need to sample from the posterior and back-propagate its parameter gradients. It can be verified that the Cholesky decomposition [11] of the covariance matrix $\boldsymbol{\Sigma}_{ij} = \boldsymbol{L}_{ij} \boldsymbol{L}_{ij}^\top$ of $q_1(\boldsymbol{z}_i, \boldsymbol{z}_j | \boldsymbol{x}_i, \boldsymbol{x}_j)$ in (4) takes the form

$$\boldsymbol{L}_{ij} = \begin{bmatrix} \boldsymbol{\sigma}_i & \boldsymbol{0}_d \\ \boldsymbol{\gamma}_{ij} \boldsymbol{\sigma}_j & \sqrt{1 - \boldsymbol{\gamma}_{ij}^2} \boldsymbol{\sigma}_j \end{bmatrix}, \quad (5)$$

allowing sampling from the approximate posterior in (4) via

$$[\boldsymbol{z}_i; \boldsymbol{z}_j] = [\boldsymbol{\mu}_i; \boldsymbol{\mu}_j] + \boldsymbol{L}_{ij} \boldsymbol{\epsilon}, \quad \text{where } \boldsymbol{\epsilon} \sim \mathcal{N}(\boldsymbol{0}_{2d}, \boldsymbol{I}_{2d}), \quad (6)$$

where $[\cdot; \cdot]$ denotes concatenation. This isolates the stochasticity in the sampling process and enables easy back-propagation of the parameter gradients from the likelihood term $\log p_{\boldsymbol{\theta}}(\boldsymbol{x}_i, \boldsymbol{x}_j | \boldsymbol{z}_i, \boldsymbol{z}_j)$ without special treatment. Further, after some algebraic manipulations, the KL-term between the homophilic posterior and prior can be derived in closed-form, omitted here for brevity (see the SM). This gives us the ELBO of the VHE with complete observations as follows

$$\mathcal{L}_{\boldsymbol{\theta}, \phi}(\boldsymbol{x}_i, \boldsymbol{x}_j | w_{ij}) = w_{ij} \left( \mathbb{E}_{\boldsymbol{z}_i, \boldsymbol{z}_j \sim q_1(\boldsymbol{z}_i, \boldsymbol{z}_j)} [\log p_{\boldsymbol{\theta}}(\boldsymbol{x}_i, \boldsymbol{x}_j | \boldsymbol{z}_i, \boldsymbol{z}_j)] - \text{KL}(q_1(\boldsymbol{z}_i, \boldsymbol{z}_j) \| p_1(\boldsymbol{z}_i, \boldsymbol{z}_j)) \right) \quad (7)$$
$$+ (1 - w_{ij}) \left( \mathbb{E}_{\boldsymbol{z}_i, \boldsymbol{z}_j \sim q_0(\boldsymbol{z}_i, \boldsymbol{z}_j)} [\log p_{\boldsymbol{\theta}}(\boldsymbol{x}_i, \boldsymbol{x}_j | \boldsymbol{z}_i, \boldsymbol{z}_j)] - \text{KL}(q_0(\boldsymbol{z}_i, \boldsymbol{z}_j) \| p_0(\boldsymbol{z}_i, \boldsymbol{z}_j)) \right).$$

**Learning with incomplete edge observations** In real-world scenarios, complete vertex information may not be available. To allow for incomplete edge observations, we also consider (where necessary) $w_{ij}$ as latent variables that need to be inferred, and define the prior for pairs without corresponding edge information as $\tilde{p}(\boldsymbol{z}_i, \boldsymbol{z}_j, w_{ij}) = p(\boldsymbol{z}_i, \boldsymbol{z}_j | w_{ij}) p(w_{ij})$, where $w_{ij} \sim \mathcal{B}(\pi_0)$ is a Bernoulli variable with parameter $\pi_0$, which can be fixed based on prior knowledge or estimated from data. For inference, we use the following approximate posterior

$$\tilde{q}(\boldsymbol{z}_i, \boldsymbol{z}_j, w_{ij} | \boldsymbol{x}_i, \boldsymbol{x}_j) = q(\boldsymbol{z}_i, \boldsymbol{z}_j | \boldsymbol{x}_i, \boldsymbol{x}_j, w_{ij}) q(w_{ij} | \boldsymbol{x}_i, \boldsymbol{x}_j), \quad (8)$$

where $q(w_{ij}|\boldsymbol{x}_i, \boldsymbol{x}_j) = \mathcal{B}(\pi_{ij})$ with $\pi_{ij} \in [0, 1]$, a neural-network-based function of the paired input $\{\boldsymbol{x}_i, \boldsymbol{x}_j\}$. Note that by integrating out $w_{ij}$, the approximate posterior for $\{\boldsymbol{z}_i, \boldsymbol{z}_j\}$ is a mixture of two Gaussians, and its corresponding ELBO is detailed in the SM, denoted as

$$\tilde{\mathcal{L}}_{\theta,\phi}(\boldsymbol{x}_i, \boldsymbol{x}_j) = \mathbb{E}_{\tilde{q}_\phi(\boldsymbol{z}_i, \boldsymbol{z}_j, w_{ij}|\boldsymbol{x}_i, \boldsymbol{x}_j)}[\log p_\theta(\boldsymbol{x}_i, \boldsymbol{x}_j|\boldsymbol{z}_i, \boldsymbol{z}_j, w_{ij})] - \mathrm{KL}(\tilde{q}_\phi(\boldsymbol{z}_i, \boldsymbol{z}_j, w_{ij}|\boldsymbol{x}_i, \boldsymbol{x}_j) \parallel \tilde{p}(\boldsymbol{z}_i, \boldsymbol{z}_j, w_{ij})). \quad (9)$$

**VHE training**   Let $\mathcal{D}_o = \{\boldsymbol{x}_i, \boldsymbol{x}_j, w_{ij}|w_{ij} \in \{0, 1\}\}$ and $\mathcal{D}_u = \{\boldsymbol{x}_i, \boldsymbol{x}_j, w_{ij}|w_{ij} = \text{N/A}\}$ be the set of complete and incomplete observations, respectively. Our training objective can be written as

$$\mathcal{L}_{\theta,\phi} = \sum_{\{\boldsymbol{x}_i, \boldsymbol{x}_j, w_{ij}\} \in \mathcal{D}_o} \mathcal{L}_{\theta,\phi}(\boldsymbol{x}_i, \boldsymbol{x}_j|w_{ij}) + \sum_{\{\boldsymbol{x}_i, \boldsymbol{x}_j\} \in \mathcal{D}_u} \tilde{\mathcal{L}}_{\theta,\phi}(\boldsymbol{x}_i, \boldsymbol{x}_j). \quad (10)$$

In practice, it is difficult to distinguish vertices with no link from those with a missing link. Hence, we propose to randomly drop a small portion, $\alpha \in [0, 1]$, of the edges with complete observations, thus treating their corresponding vertex pairs as incomplete observations. Empirically, this uncertainty modeling improves model robustness, boosting performance. Following standard practice, we draw mini-batches of data and use stochastic gradient ascent to learn model parameters $\{\theta, \phi\}$ with (10).

## 3.2   VHE for networks with textual attributes

In this section, we provide implementation details of VHE on textual networks as a concrete example.

**Encoder architecture**   The schematic diagram of our VHE encoder for textual networks is provided in Figure 2. Our encoder design utilizes $(i)$ a novel phrase-to-word alignment-based text embedding module to extract context-dependent features from vertex text; $(ii)$ a lookup-table-based structure embedding to capture topological features of vertices; and $(iii)$ a neural integrator that combines semantic and topological features to infer the approximate posterior of latent codes.

*Phrase-to-word alignment:* Given the associated text on a pair of vertices, $\boldsymbol{x}_i \in \mathbb{R}^{d_w \times L_i}$ and $\boldsymbol{x}_j \in \mathbb{R}^{d_w \times L_j}$, where $d_w$ is the dimension of word embeddings, and $L_i$ and $L_j$ are the length of each text sequence. We treat $\boldsymbol{x}_j$ as the context of $\boldsymbol{x}_i$, and *vice versa*. Specifically, we first compute token-wise similarity matrix $\boldsymbol{M} = \boldsymbol{x}_i^T \boldsymbol{x}_j \in \mathbb{R}^{L_i \times L_j}$. Next, we compute row-wise and column-wise weight vectors based on $\boldsymbol{M}$ to aggregate features for $\boldsymbol{x}_i$ and $\boldsymbol{x}_j$. To this end, we perform 1D convolution on $\boldsymbol{M}$ both row-wise and column-wise, followed by a $\tanh(\cdot)$ activation, to capture phrase-to-word

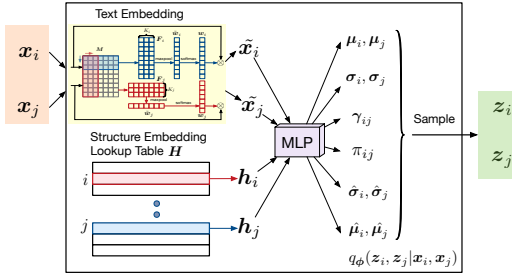

Figure 2: Illustration of the proposed VHE encoder. See SM for a larger version of the text embedding module.

similarities. This results in $\boldsymbol{F}_i \in \mathbb{R}^{L_i \times K_r}$ and $\boldsymbol{F}_j \in \mathbb{R}^{L_j \times K_c}$, where $K_r$ and $K_c$ are the number of filters for rows and columns, respectively. We then aggregate via max-pooling on the second dimension to combine the convolutional outputs, thus collapsing them into 1D arrays, *i.e.*, $\tilde{\boldsymbol{w}}_i = \text{max-pool}(\boldsymbol{F}_i)$ and $\tilde{\boldsymbol{w}}_j = \text{max-pool}(\boldsymbol{F}_j)$. After softmax normalization, we have the phrase-to-word alignment vectors $\boldsymbol{w}_i \in \mathbb{R}^{L_i}$ and $\boldsymbol{w}_j \in \mathbb{R}^{L_j}$. The final text embeddings are given by $\tilde{\boldsymbol{x}}_i = \boldsymbol{x}_i \boldsymbol{w}_i \in \mathbb{R}^{d_w}$, and similarly for $\tilde{\boldsymbol{x}}_j$. Additional details are provided in the SM.

*Structure embedding and neural integrator:* For each vertex $v_i$, we assign a $d_w$-dimensional learnable parameter $\boldsymbol{h}_i$ as structure embedding, which seeks to encode the topological information of the vertex. The set of all structure embeddings $\boldsymbol{H}$ constitutes a look-up table for all vertices in $\boldsymbol{G}$. The aligned text embeddings and structure embeddings are concatenated into a feature vector $\boldsymbol{f}_{ij} \triangleq [\tilde{\boldsymbol{x}}_i; \tilde{\boldsymbol{x}}_j; \boldsymbol{h}_i; \boldsymbol{h}_j] \in \mathbb{R}^{4d_w}$, which is fed into the neural integrator to obtain the posterior means ($\boldsymbol{\mu}_i, \boldsymbol{\mu}_j$, $\hat{\boldsymbol{\mu}}_i, \hat{\boldsymbol{\mu}}_j$), covariance ($\boldsymbol{\sigma}_i^2, \boldsymbol{\sigma}_j^2, \hat{\boldsymbol{\sigma}}_i^2, \hat{\boldsymbol{\sigma}}_j^2$) and homophily factors $\boldsymbol{\gamma}_{ij}$. For pairs with missing edge information, *i.e*, $w_{ij} = \text{N/A}$, the neural integrator also outputs the posterior probability of edge presence, *i.e.*, $\pi_{ij}$. A standard multi-layer perceptron (MLP) is used for the neural integrator.

**Decoder architecture**   Key to the design of the decoder is the specification of a conditional likelihood model from a latent code $\{\boldsymbol{z}_i, \boldsymbol{z}_j\}$ to an observation $\{\boldsymbol{x}_i, \boldsymbol{x}_j\}$. Two choices can be considered: $(i)$ direct reconstruction of the original text sequence (conditional multinomial likelihood), and $(ii)$ indirect reconstruction of the text sequence in embedding space (conditional Gaussian likelihood). In practice, the direct approach typically also encodes irrelevant nuisance information [18], thus we follow the indirect approach. More specifically, we use the max-pooling feature

$$\mathring{\boldsymbol{x}}_i = \text{max-pool}(\boldsymbol{x}_i), \ \ \mathring{\boldsymbol{x}}_j = \text{max-pool}(\boldsymbol{x}_j), \quad (11)$$

as the target representation, and let

$$\log p_\theta(\boldsymbol{x}_i, \boldsymbol{x}_j | \boldsymbol{z}_i, \boldsymbol{z}_j) = -(||\mathring{\boldsymbol{x}}_i - \hat{\boldsymbol{x}}_i(\boldsymbol{z}_i; \theta)||^2 + ||\mathring{\boldsymbol{x}}_j - \hat{\boldsymbol{x}}_j(\boldsymbol{z}_j; \theta)||^2), \qquad (12)$$

where $\hat{\boldsymbol{x}}(\boldsymbol{z}; \theta) = f_\theta(\boldsymbol{z})$, is the reconstruction of $\mathring{\boldsymbol{x}}$ by passing posterior sample $\boldsymbol{z}$ through MLP $f_\theta(\boldsymbol{z})$.

### 3.3 Inference at test time

**Global network embedding** Above we have defined *localized* vertex embedding of $v_i$ by conditioning on another vertex $v_j$ (the context). For many network learning tasks, a *global* vertex embedding is desirable, *i.e.*, without conditioning on any specific vertex. To this end, we identify the global vertex embedding distribution by simply averaging all the pairwise local embeddings $p_\phi(\boldsymbol{z}_i | \boldsymbol{X}) = \frac{1}{N-1} \sum_{j \neq i} q(\boldsymbol{z}_i | \boldsymbol{x}_i, \boldsymbol{x}_j, w_{ij})$, where, with a slight abuse of notation, $q(\boldsymbol{z}_i | \boldsymbol{x}_i, \boldsymbol{x}_j, w_{ij})$ denotes the approximate posterior both with and without edge information ($\tilde{q}$ in (8)). In this study, we summarize the distribution via expectations, *i.e.*, $\bar{\boldsymbol{z}}_i = \mathbb{E}[p_\phi(\boldsymbol{z}_i | \boldsymbol{X})] = \frac{1}{N-1} \sum_{j \neq i} \mathbb{E}[q(\boldsymbol{z}_i | \boldsymbol{x}_i, \boldsymbol{x}_j, w_{ij})] \in \mathbb{R}^d$, which can be computed in closed form from (4). For large-scale networks, where the exact computation of $\bar{\boldsymbol{z}}_i$ is computationally unfeasible, we use Monte Carlo estimates by subsampling $\{\boldsymbol{x}_j\}_{j \neq i}$.

**Generalizing to unseen vertices** Unlike most existing approaches, our generative formulation generalizes to unseen vertices. Assume we have a model with learned parameters $(\hat{\theta}, \hat{\phi})$, and learned structure-embedding $\hat{\boldsymbol{H}}$ of vertices in the training set, hereafter collectively referred to as $\hat{\Theta}$. For an unseen vertex $v_\star$ with associated text data $\boldsymbol{x}_\star$, we can learn its structure-embedding $\boldsymbol{h}_\star$ by optimizing it to maximize the average of variational bounds with the learned (global) parameters fixed, *i.e.*, $\mathcal{J}(\boldsymbol{h}_\star) \triangleq \frac{1}{N} \sum_{\boldsymbol{x}_i \in \boldsymbol{X}} \tilde{\mathcal{L}}_{\hat{\theta}, \hat{\phi}}(\boldsymbol{x}_\star, \boldsymbol{x}_i | \boldsymbol{h}_\star, \hat{\boldsymbol{H}})$. Then the inference method above can be reused with the optimized $\boldsymbol{h}_\star$ to obtain $p_\phi(\boldsymbol{z}_\star | \boldsymbol{X})$.

## 4  Related Work

**Network Embedding** Classical network embedding schemes mainly focused on the preservation of network topology (*e.g.,* edges). For example, early developments explored direct low-rank factorization of the affinity matrix [1] or its Laplacian [6]. Alternative to such deterministic graph factorization solutions, models such as LINE [37] employed a probabilistic formulation to account for the uncertainty of edge information. Motivated by the success of word embedding in NLP, DeepWalk [31] applied the skip-gram model to the sampled diffusion paths, capturing the local interactions between the vertices. More generally, higher-level properties such as community structure can also be preserved with specific embedding schemes [47]. Generalizations to these schemes include Node2Vec [16], UltimateWalk [9], amongst others. Despite their wide-spread empirical success, limitations of such topology-only embeddings have also been recognized. In real-world scenario, the observed edges are usually sparse relative to the number of all possible interactions, and substantial measurement error can be expected, violating the working assumptions of these models [30]. Additionally, these solutions typically cannot generalize to unseen vertices.

Fortunately, apart from the topology information, many real-world networks are also associated with rich side information (*e.g.*, labels, texts, attributes, images, etc.), commonly known as attributes, on each vertex. The exploration of this additional information, together with the topology-based embedding, has attracted recent research interest. For example, this can be achieved by accounting for the explicit vertex labels [41], or by modeling the latent topics of the vertex content [38]. Alternatively, [49] learns a topology-preserving embedding of the side information to factorize the DeepWalk diffusion matrix. To improve the flexibility of fixed-length embeddings, [40] instead treats side information as *context* and advocates the use of *context-aware network embeddings* (CANE). From the theoretical perspective, with additional technical conditions, formal inference procedures can be established for such context-dependent embeddings, which guarantees favorable statistical properties such as uniform consistency and asymptotic normality [48]. CANE has also been further improved by using more fine-grained word-alignment approaches [34].

Notably, all methods discussed above have almost exclusivley focused on the use of discriminative objectives. Compared with them, we presents a novel, fully generative model for summarizing both the topological and semantic information of a network, which shows superior performance for link prediction, and better generalization capabilities to unseen vertices (see the Experiments section).

**Variational Autoencoder** VAE [21] is a powerful framework for learning stochastic representations that account for model uncertainty. While its applications have been extensively studied in the

context of computer vision and NLP [44, 33, 50], its use in complex network analysis is less widely explored. Existing solutions focused on building VAEs for the generation of a graph, but not the associated contents [22, 35, 26]. Such practice amounts to a variational formulation of a discriminative goal, thereby compromising more general downstream network learning tasks. To overcome such limitations, we model pairwise data rather than a singleton as in standard VAE. Recent literature has also started to explore priors other than standard Gaussian to improve model flexibility [12, 39, 46, 45], or enforce structural knowledge [2]. In our case, we have proposed a novel homophlic prior to exploit the correlation in the latent representation of connected vertices.

## 5 Experiments

We evaluate the proposed VHE on link prediction and vertex classification tasks. Our code is available from https://github.com/Wenlin-Wang/VHE19.

**Datasets** Following [40], we consider three widely studied real-world network datasets: CORA [28], HEPTH [25], and ZHIHU[1]. CORA and HEPTH are citation network datasets, and ZHIHU is a network derived from the largest $Q\&A$ website in China. Summary statistics for these datasets are provided in Table 1. To make direct comparison with existing work, we adopt the same pre-processing steps described in [40, 34]. Details of the experimental setup are found in the SM.

Table 1: Summary of datasets used in evaluation.

| Datasets | #vertices | #edges | %sparsity | #labels |
|---|---|---|---|---|
| CORA | 2,277 | 5,214 | 0.10% | 7 |
| HEPTH | 1,038 | 1,990 | 0.18% | – |
| ZHIHU | 10,000 | 43,894 | 0.04% | – |

**Evaluation metrics** AUC score [17] is employed as the evaluation metric for link prediction. For vertex classification, we follow [49] and build a linear SVM [14] on top of the learned network embedding to predict the label for each vertex. Various training ratios are considered, and for each, we repeat the experiment 10 times and report the mean score and the standard deviation.

**Baselines** To demonstrate the effectiveness of VHE, three groups of network embedding approaches are considered: (*i*) *Structural*-based methods, including MMB [3], LINE [37], Node2Vec [16] and DeepWalk [31]. (*ii*) Approaches that utilize both *structural* and *semantic* information, including TADW [49], CENE [36], CANE [40] and WANE [34]. (*iii*) VAE-based generative approaches, including the naïve variational solution discussed in Section 3, using Node2Vec [16] as the off-the-shelf structural embedding (Naïve-VAE), and VGAE [22]. To verify the effectiveness of the proposed generative objective and phrase-to-word alignment, a baseline model employing the same textual embedding as VHE, but with a discriminative objective [40], is also considered, denoted as PWA. For vertex classification, we further compare with DMTE [51].

### 5.1 Results and analysis

Table 2: AUC scores for link prediction on three benchmark datasets. Each experiment is repeated 10 times, and the standard deviation is found in the SM, together with more detailed results.

| Data | CORA | | | | | HEPTH | | | | | ZHIHU | | | | |
|---|---|---|---|---|---|---|---|---|---|---|---|---|---|---|---|
| % Train Edges | 15% | 35% | 55% | 75% | 95% | 15% | 35% | 55% | 75% | 95% | 15% | 35% | 55% | 75% | 95% |
| MMB [3] | 54.7 | 59.5 | 64.9 | 71.1 | 75.9 | 54.6 | 57.3 | 66.2 | 73.6 | 80.3 | 51.0 | 53.7 | 61.6 | 68.8 | 72.4 |
| LINE [37] | 55.0 | 66.4 | 77.6 | 85.6 | 89.3 | 53.7 | 66.5 | 78.5 | 87.5 | 87.6 | 52.3 | 59.9 | 64.3 | 67.7 | 71.1 |
| Node2Vec [16] | 55.9 | 66.1 | 78.7 | 85.9 | 88.2 | 57.1 | 69.9 | 84.3 | 88.4 | 89.2 | 54.2 | 57.3 | 58.7 | 66.2 | 68.5 |
| DeepWalk [31] | 56.0 | 70.2 | 80.1 | 85.3 | 90.3 | 55.2 | 70.0 | 81.3 | 87.6 | 88.0 | 56.6 | 60.1 | 61.8 | 63.3 | 67.8 |
| TADW [49] | 86.6 | 90.2 | 90.0 | 91.0 | 92.7 | 87.0 | 91.8 | 91.1 | 93.5 | 91.7 | 52.3 | 55.6 | 60.8 | 65.2 | 69.0 |
| CENE [36] | 72.1 | 84.6 | 89.4 | 93.9 | 95.9 | 86.2 | 89.8 | 92.3 | 93.2 | 93.2 | 56.8 | 60.3 | 66.3 | 70.2 | 73.8 |
| CANE [40] | 86.8 | 92.2 | 94.6 | 95.6 | 97.7 | 90.0 | 92.0 | 94.2 | 95.4 | 96.3 | 56.8 | 62.9 | 68.9 | 71.4 | 75.4 |
| WANE [34] | 91.7 | 94.1 | 96.2 | 97.5 | 99.1 | 92.3 | 95.7 | 97.5 | 97.7 | 98.7 | 58.7 | 68.3 | 74.9 | 79.7 | 82.6 |
| Naïve-VAE | 60.2 | 67.8 | 80.2 | 87.7 | 90.1 | 60.8 | 68.1 | 80.7 | 88.8 | 90.5 | 56.5 | 60.2 | 62.5 | 68.1 | 69.0 |
| VGAE [22] | 63.9 | 74.3 | 84.3 | 88.1 | 90.5 | 65.5 | 74.5 | 85.9 | 88.4 | 90.4 | 55.9 | 61.9 | 64.6 | 70.1 | 71.2 |
| PWA | 92.2 | 95.6 | 96.8 | 97.7 | 98.9 | 92.8 | 96.1 | 97.6 | 97.9 | 99.0 | 62.6 | 70.8 | 77.1 | 80.8 | 83.3 |
| VHE | **94.4** | **97.6** | **98.3** | **99.0** | **99.4** | **94.1** | **97.5** | **98.3** | **98.8** | **99.4** | **66.8** | **74.1** | **81.6** | **84.7** | **86.4** |

**Link Prediction** Given the network, various ratios of observed edges are used for training and the rest are used for testing. The goal is to predict missing edges. Results are summarized in Table 2 (more comprehensive results can be found in the SM). One may make several observations: (*i*) Semantic-aware methods are consistently better than approaches that only use structural information, indicating the importance of incorporating associated text sequences into network embeddings. (*ii*)

When comparing PWA with CANE [40] and WANE [34], it is observed that the proposed phrase-to-word alignment performs better than other competing textual embedding methods. (*iii*) Naïve VAE solutions are less effective. Semantic information extracted from standard VAE (Naïve-VAE) provides only incremental improvements relative to structure-only approaches. VGAE [22] neglects the semantic information, and cannot scale to large datasets (its performance is also subpar). (*iv*) VHE achieves consistently superior performance on all three datasets across different missing-data levels, which suggests that VHE is an effective solution for learning network embeddings, especially when the network is sparse and large. As can be seen from the largest dataset ZHIHU, VHE achieves an average of 5.9 points improvement in AUC score relative to the prior state-of-the-art, WANE [34].

**Vertex Classification** The effectiveness of the learned network embedding is further investigated on vertex classification. Similar to [40], learned embeddings are saved and then a SVM is built to predict the label for each vertex. Both quantitative and qualitative results are provided, with the former shown in Table 3. Similar to link prediction, semantic-aware approaches, *e.g.*, CENE [36], CANE [40], WANE [34], provide better performance than structure-only approaches. Furthermore, VHE outperforms other strong baselines as well as our PWA model, indicating that VHE is capable of best leveraging the structural and semantic information, resulting in robust network embed-

Table 3: Test accuracy for vertex classification on the CORA dataset.

| % of Labeled Data | 10% | 30% | 50% | 70% |
|---|---|---|---|---|
| DeepWalk [31] | 50.8 | 54.5 | 56.5 | 57.7 |
| LINE [37] | 53.9 | 56.7 | 58.8 | 60.1 |
| CANE [40] | 81.6 | 82.8 | 85.2 | 86.3 |
| TADW [49] | 71.0 | 71.4 | 75.9 | 77.2 |
| WANE [34] | 81.9 | 83.9 | 86.4 | 88.1 |
| DMTE [51] | 81.8 | 83.9 | 86.4 | 88.1 |
| PWA | 82.1 | 83.8 | 86.7 | 88.2 |
| VHE | **82.6** | **84.3** | **87.7** | **88.5** |

dings. As qualitative analysis, we use *t*-SNE [27] to visualize the learned embeddings, as shown in Figure 4(a). Vertices belonging to different classes are well separated from each other.

**When does VHE works?** VHE produces state-of-the-art results, and an additional question concerns analysis of when our VHE works better than previous discriminative approaches. Intuitively, VHE imposes strong *structural* constraints, and could add to more robust estimation, especially when the vertex connections are sparse. To validate this hypothesis, we design the following experiment on ZHIHU. When evaluating the model, we separate the testing vertices into quantiles based on the number of edges of each vertex (degree), to compare VHE against PWA on each group. Results are summarized in Figure 3. VHE improves link prediction for all groups of vertices, and the gain is large especially when the interactions between

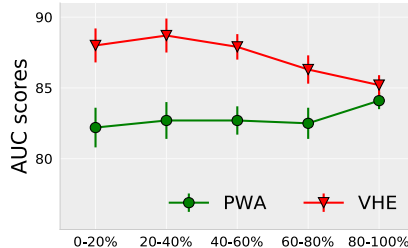

Figure 3: AUC as a function of vertex degree (quantiles). Error bars represent the standard deviation.

vertices are rare, evidence that our proposed *structural* prior is a rational assumption and provides robust learning of network embeddings. Also interesting is that prediction accuracy on groups with rare connections is no worse than those with dense connections. One possible explanation is that the semantic information associated with the group of users with rare connections is more related to their true interests, hence it can be used to infer the connections accurately, while such semantic information could be noisy for those active users with dense connections.

**Link prediction on unseen vertices** VHE can be further extended for learning embeddings for unseen vertices, which has not been well studied previously. To investigate this, we split the vertices into training/testing sets with various ratios, and report the link prediction results on those unseen (testing) vertices. To evaluate the generalization ability of previous discriminative approaches to unseen vertices, two variants of CANE [40] and WANE [34] are considered as our baselines. (*i*) The first method ignores the structure embedding and purely relies on the semantic textual information to infer the edges, and therefore can be directly extended for unseen vertices (marked by †). (*ii*) The second approach learns an additional mapping from the semantic embedding to the structure embedding with an MLP during training. When testing on unseen vertices, it first infers the structure embedding from its semantic embedding, and then combines with the semantic embedding to predict the existence of links (marked by ‡). Results are provided in Table 4. Consistent with previous results, semantic information is useful for link prediction. Though the number of vertices we observe is small, *e.g.*, 15%, VHE and other semantic-aware approaches can predict the links reasonably well. Further, VHE consistently outperforms PWA, showing that the proposed variational approach used in VHE yields better generalization performance for unseen vertices than discriminative models.

Table 4: AUC scores under the setting with unseen vertices for link prediction. † denotes approaches using semantic features only, ‡ denotes methods using both semantic and structure features, and the structure features are inferred from the semantic features with a one-layer MLP.

| Data | CORA | | | | | HEPTH | | | | | ZHIHU | | | | |
|---|---|---|---|---|---|---|---|---|---|---|---|---|---|---|---|
| % Train Vertices | 15% | 35% | 55% | 75% | 95% | 15% | 35% | 55% | 75% | 95% | 15% | 35% | 55% | 75% | 95% |
| CANE† | 83.4 | 87.9 | 91.1 | 93.8 | 95.1 | 84.2 | 88.0 | 91.2 | 93.6 | 94.7 | 55.9 | 62.1 | 67.3 | 73.3 | 76.2 |
| CANE‡ | 83.1 | 86.8 | 90.4 | 93.9 | 95.2 | 83.8 | 88.0 | 91.0 | 93.7 | 95.0 | 56.0 | 61.5 | 66.9 | 73.5 | 76.3 |
| WANE† | 87.4 | 88.7 | 92.2 | 94.2 | 95.7 | 86.6 | 88.4 | 92.8 | 93.8 | 95.2 | 57.6 | 65.1 | 71.2 | 76.6 | 79.9 |
| WANE‡ | 87.0 | 88.8 | 92.5 | 95.4 | 95.7 | 86.9 | 88.3 | 92.8 | 94.1 | 95.3 | 57.8 | 65.2 | 70.8 | 76.5 | 80.2 |
| PWA† | 87.7 | 89.9 | 93.5 | 95.7 | 95.9 | 87.2 | 90.2 | 93.1 | 95.2 | 96.1 | 61.5 | 74.7 | 77.3 | 81.0 | 82.3 |
| PWA‡ | 87.8 | 90.1 | 93.3 | 95.8 | 96.0 | 87.4 | 90.5 | 93.0 | 95.5 | 96.2 | 62.0 | 75.0 | 77.4 | 80.9 | 82.4 |
| VHE | **89.9** | **92.4** | **95.0** | **96.9** | **97.4** | **90.2** | **92.6** | **94.8** | **96.6** | **97.7** | **63.2** | **75.6** | **78.0** | **81.3** | **82.7** |

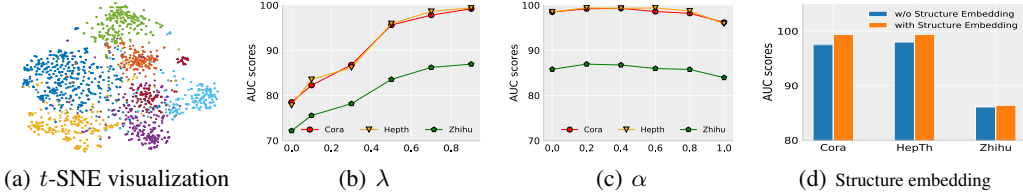

(a) $t$-SNE visualization  (b) $\lambda$  (c) $\alpha$  (d) Structure embedding

Figure 4: (a) $t$-SNE visualization of the learned network embedding on the CORA dataset, labels are color coded. (b, c) Sensitivity analysis of hyper-parameter $\lambda$ and $\alpha$, respectively. (d) Ablation study on the use of structure embedding in the encoder. Results are reported using 95% training edges on the three datasets.

## 5.2 Ablation study

**Sensitivity analysis** The homophily factor $\lambda$ controls the strength of the linking information. To analyze its impact on the performance of VHE, we conduct experiments with 95% training edges on the CORA dataset. As observed from Figure 4(b), empirically, a larger $\lambda$ is preferred. This is intuitive, since the ultimate goal is to predict structural information, and our VHE incorporates such information in the prior design. If $\lambda$ is large, the structural information plays a more important role in the objective, and the optimization of the ELBO in (7) will seek to accommodate such information. It is also interesting to note that VHE performs well even when $\lambda = 0$. In this case, embeddings are purely inferred from the semantic features learned from our model, and such semantic information may have strong correlations with the structure information.

In Figure 4(c), we further investigate the sensitivity of our model to the dropout ratio $\alpha$. With a small dropout ratio ($0 < \alpha \leq 0.4$), we observe consistent improvements over the no drop-out baseline ($\alpha = 0$) across all datasets, demonstrating the effectiveness of uncertainty estimation for link prediction. Even when the dropout ratio is $\alpha = 1.0$, the performance does not drop dramatically. We hypothesize that this is because the VHE is able to discover the underlying missing edges given our homophilic prior design.

**Structure embedding** Our encoder produces both *semantic* and *structure*-based embeddings for each vertex. We analyze the impact of the structure embedding. Experiments with and without structure embeddings are performed on the three datasets. Results are shown in Figure 4(d). We find that without the structure embedding, the performance remains almost the same for the ZHIHU dataset. However, the AUC scores drops about 2 points for the other two datasets. It appears that the impact of the structure embedding may vary across datasets. The semantic information in CORA and HEPTH may not fully reflect its structure information, *e.g.*, documents with similar semantic information are not necessary to cite each other.

## 6 Conclusions

We have presented Variational Homophilic Embedding (VHE), a novel method to characterize relationships between vertices in a network. VHE learns informative and robust network embeddings by leveraging *semantic* and *structural* information. Additionally, a powerful phrase-to-word alignment approach is introduced for textual embedding. Comprehensive experiments have been conducted on link prediction and vertex-classification tasks, and state-of-the-art results are achieved. Moreover, we provide insights for the benefits brought by VHE, when compared with traditional discriminative models. It is of interest to investigate the use of VHE in more complex scenarios, such as learning node embeddings for graph matching problems.

**Acknowledgement**: This research was supported in part by DARPA, DOE, NIH, ONR and NSF.

## Footnotes

[1]https://www.zhihu.com/

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
