[Supplementary Material]

# A  Derivation of the ELBO

Recall that we have

$$p_1 = p_1(\boldsymbol{z}_i, \boldsymbol{z}_j) = \mathcal{N}\left( \begin{bmatrix} \mathbf{0}_d \\ \mathbf{0}_d \end{bmatrix}, \begin{bmatrix} \boldsymbol{I}_d & \lambda \cdot \boldsymbol{I}_d \\ \lambda \cdot \boldsymbol{I}_d & \boldsymbol{I}_d \end{bmatrix} \right), \, p_0 = p_0(\boldsymbol{z}_i, \boldsymbol{z}_j) = \mathcal{N}\left( \begin{bmatrix} \mathbf{0}_d \\ \mathbf{0}_d \end{bmatrix}, \begin{bmatrix} \boldsymbol{I}_d & \mathbf{0}_d \\ \mathbf{0}_d & \boldsymbol{I}_d \end{bmatrix} \right), \quad (13)$$

for the prior distribution, and similarly, we have the approximate posterior as below:

$$q_1 = q_1(\boldsymbol{z}_i, \boldsymbol{z}_j) = \mathcal{N}\left( \begin{bmatrix} \boldsymbol{\mu}_i \\ \boldsymbol{\mu}_j \end{bmatrix}, \begin{bmatrix} \boldsymbol{\sigma}_i^2 & \boldsymbol{\gamma}_{ij}\boldsymbol{\sigma}_i\boldsymbol{\sigma}_j \\ \boldsymbol{\gamma}_{ij}\boldsymbol{\sigma}_i\boldsymbol{\sigma}_j & \boldsymbol{\sigma}_j^2 \end{bmatrix} \right), \, q_0 = q_0(\boldsymbol{z}_i, \boldsymbol{z}_j) = \mathcal{N}\left( \begin{bmatrix} \hat{\boldsymbol{\mu}}_i \\ \hat{\boldsymbol{\mu}}_j \end{bmatrix}, \begin{bmatrix} \hat{\boldsymbol{\sigma}}_i^2 & \mathbf{0}_d \\ \mathbf{0}_d & \hat{\boldsymbol{\sigma}}_j^2 \end{bmatrix} \right). \quad (14)$$

Now, we derive the ELBO ($\tilde{\mathcal{L}}_{\theta,\phi}(\boldsymbol{x}_i, \boldsymbol{x}_j)$) when observing incomplete edges as follow:

$$\tilde{\mathcal{L}}_{\theta,\phi}(\boldsymbol{x}_i, \boldsymbol{x}_j) = \mathbb{E}_{\tilde{q}_\phi(\boldsymbol{z}_i, \boldsymbol{z}_j, w_{ij}|\boldsymbol{x}_i, \boldsymbol{x}_j)}[\log p_\theta(\boldsymbol{x}_i, \boldsymbol{x}_j|\boldsymbol{z}_i, \boldsymbol{z}_j, w_{ij})] \quad (15)$$

$$- \mathrm{KL}(\tilde{q}_\phi(\boldsymbol{z}_i, \boldsymbol{z}_j, w_{ij}|\boldsymbol{x}_i, \boldsymbol{x}_j) \parallel \tilde{p}(\boldsymbol{z}_i, \boldsymbol{z}_j, w_{ij})). \quad (16)$$

Specifically, $\tilde{p}(\boldsymbol{z}_i, \boldsymbol{z}_j, w_{ij}) = p(\boldsymbol{z}_i, \boldsymbol{z}_j|w_{ij})p(w_{ij})$, where $p(w_{ij}) = \mathcal{B}(\pi_0)$ is a Bernoulli variable with parameter $\pi_0$. When integrating $w_{ij}$ out, we have $\tilde{p}(\boldsymbol{z}_i, \boldsymbol{z}_j) = \pi_0 \cdot p_1 + (1 - \pi_0) \cdot p_0$, which is essentially a mixture of two Gaussians, Similarly, the approximated posterior is $\tilde{q}_\phi(\boldsymbol{z}_i, \boldsymbol{z}_j, w_{ij}) = q(\boldsymbol{z}_i, \boldsymbol{z}_j|w_{ij})q(w_{ij})$, where $q_\phi(w_{ij}) = \mathcal{B}(\pi_{ij})$. When integrating $w_{ij}$ out, we have $q(\boldsymbol{z}_i, \boldsymbol{z}_j) = \pi_{ij} \cdot q_1 + (1 - \pi_{ij}) \cdot q_0$.

In order to optimize the corresponding ELBO, we need the KL divergence between the inferred posterior and the proposed prior. A detailed derivation for the KL divergence is as follows:

$$\mathrm{KL}(\tilde{q}_\phi(\boldsymbol{z}_i, \boldsymbol{z}_j, w_{ij}) \parallel p(\boldsymbol{z}_i, \boldsymbol{z}_j, w_{ij})) = \int_{\boldsymbol{z}_i, \boldsymbol{z}_j} \sum_{w_{ij}} \tilde{q}_\phi(\boldsymbol{z}_i, \boldsymbol{z}_j, w_{ij}) \log \frac{\tilde{q}_\phi(\boldsymbol{z}_i, \boldsymbol{z}_j, w_{ij})}{\tilde{p}(\boldsymbol{z}_i, \boldsymbol{z}_j, w_{ij})} d\boldsymbol{z}_i d\boldsymbol{z}_j$$

$$= \int_{\boldsymbol{z}_i, \boldsymbol{z}_j} \sum_{w_{ij}} q_\phi(\boldsymbol{z}_i, \boldsymbol{z}_j|w_{ij})q_\phi(w_{ij}) \log \frac{q_\phi(\boldsymbol{z}_i, \boldsymbol{z}_j|w_{ij})q_\phi(w_{ij})}{p(\boldsymbol{z}_i, \boldsymbol{z}_j|w_{ij})p(w_{ij})} d\boldsymbol{z}_i d\boldsymbol{z}_j$$

$$= \int \pi_{ij} \cdot q_1 \log \frac{q_1}{p_1} + \int (1 - \pi_{ij}) \cdot q_0 \log \frac{q_0}{p_0} + \sum_{w_{ij}} q_\phi(w_{ij}) \log \frac{q_\phi(w_{ij})}{p(w_{ij})}$$

$$= \pi_{ij} \cdot \mathrm{KL}(q_1 \parallel p_1) + (1 - \pi_{ij}) \cdot \mathrm{KL}(q_0 \parallel p_0) + \mathrm{KL}(q_\phi(w_{ij}) \parallel p(w_{ij})). \quad (17)$$

Additionally, different from standard VAE [21], our prior $p_1$ and the approximate posterior $q_1$ includes correlation terms. With properties provided in [32], the KL term enjoys a closed-form expression:

$$\mathrm{KL}(q_1 \parallel p_1) = \frac{1}{2}\mathbf{1}_d^T \big[ \log(1 - \lambda^2) - \log(1 - \boldsymbol{\gamma}_{ij}^2) - \log\boldsymbol{\sigma}_i^2 - \log\boldsymbol{\sigma}_j^2 - 2$$

$$+ \frac{\boldsymbol{\sigma}_i^2 + \boldsymbol{\sigma}_j^2 - 2\lambda\boldsymbol{\gamma}_{ij}\boldsymbol{\sigma}_i\boldsymbol{\sigma}_j}{1 - \lambda^2} + \frac{\boldsymbol{\mu}_i^2 + \boldsymbol{\mu}_j^2 - 2\lambda\boldsymbol{\mu}_i\boldsymbol{\mu}_j}{1 - \lambda^2} \big]\mathbf{1}_d. \quad (18)$$

Therefore, when observing incomplete edges, $\tilde{\mathcal{L}}_{\phi,\theta}(\boldsymbol{x}_i, \boldsymbol{x}_j)$ can be readily optimized.

# B  Phrase-to-Word Alignment

Inspired by [43], we propose a phrase-by-word alignment (PWA) method to extract semantic features from the associated text. The detailed implementation is illustrated in Figure 5.

Figure 5: The detailed implementation of our phrase-to-word alignment in the encoder.

Given the associated text on a pair of vertices, $\boldsymbol{x}_i \in \mathbb{R}^{d_w \times L_i}$ and $\boldsymbol{x}_j \in \mathbb{R}^{d_w \times L_j}$, where $d_w$ is the dimension of word embeddings, and $L_i$ and $L_j$ are the length of each text sequence. We treat $\boldsymbol{x}_j$ as the context of $\boldsymbol{x}_i$, and *vice versa*. Specifically, we first compute token-wise similarity matrix $\boldsymbol{M} = \boldsymbol{x}_i^T \boldsymbol{x}_j \in \mathbb{R}^{L_i \times L_j}$. Next, we compute row-wise and column-wise weight vectors based on $\boldsymbol{M}$ to aggregate features for $\boldsymbol{x}_i$ and $\boldsymbol{x}_j$. To this end, we perform 1D convolution on $\boldsymbol{M}$ both row-wise and column-wise, followed by a $\tanh(\cdot)$ activation, to capture phrase-to-word similarities. Specifically, assuming we have $K_r$ row-wise and $K_c$ column-wise convolutional kernels, noted as $\mathbf{U} \in \mathbb{R}^{K_r \times L_j \times l}$ and $\mathbf{V} \in \mathbb{R}^{K_c \times L_i \times l}$, respectively. $l$ is the size of convolutional kernels. Then we have

$$\boldsymbol{F}_i = \tanh(\boldsymbol{M}^T \otimes \mathbf{U}) \in \mathbb{R}^{L_i \times K_r}, \quad \boldsymbol{F}_j = \tanh(\boldsymbol{M} \otimes \mathbf{V}) \in \mathbb{R}^{L_j \times K_c}, \tag{19}$$

where $\otimes$ is the 1-D convolutional operator. We then aggregate via max-pooling on the second dimension to combine the convolutional outputs, thus collapsing them into 1D arrays, defined as

$$\tilde{\boldsymbol{w}}_i = \text{max-pool}(\boldsymbol{F}_i) \in \mathbb{R}^{L_i}, \quad \tilde{\boldsymbol{w}}_j = \text{max-pool}(\boldsymbol{F}_j) \in \mathbb{R}^{L_j}. \tag{20}$$

After softmax normalization, we have the phrase-to-word alignment vectors

$$\boldsymbol{w}_i = \text{softmax}(\tilde{\boldsymbol{w}}_i) \in \mathbb{R}^{L_i}, \quad \boldsymbol{w}_j = \text{softmax}(\tilde{\boldsymbol{w}}_j) \in \mathbb{R}^{L_j}. \tag{21}$$

The final text embeddings are given by

$$\tilde{\boldsymbol{x}}_i = \boldsymbol{x}_i \boldsymbol{w}_i \in \mathbb{R}^{d_w}, \quad \tilde{\boldsymbol{x}}_j = \boldsymbol{x}_j \boldsymbol{w}_j \in \mathbb{R}^{d_w}. \tag{22}$$

# C  Experiments

## C.1  Experimental Setup

To be consistent with previous research, the embedding dimension is set to 200 for all the approaches (100 for *semantic* embedding and 100 for *structure* embedding in our model). The kernel size for our phrase-to-word alignment module is fix to 5, and 200 kernels (for both $K_i$ and $K_j$) are used. On top of the concatenation of the *semantic* and *structure* embedding, our encoder employs a one-hidden-layer MLP to infer the latent variables $(\boldsymbol{z}_i, \boldsymbol{z}_j)$, and is the same for the decoder. $\tanh$ is applied as the activation function. Adam [20] with a constant learning rate of $1 \times 10^{-4}$ is used for optimization. The hyper-parameter $\lambda, \alpha$ is fixed to 0.99 and 0.2, respectively, in all our experiments, and the sensitivity analysis of these two hyper-parameters are provided in Sec 5.2. The hyper-parameter $\pi_0$ is set to be the sparsity level for each dataset, as shown in Table 1.

## C.2  Link Prediction

The complete link-prediction results with various training edges ratios are shown in Table 5, 6, 7.

Table 5: AUC scores for link prediction on the CORA dataset.

| % Training Edges | 15% | 25% | 35% | 45% | 55% | 65% | 75% | 85% | 95% |
|---|---|---|---|---|---|---|---|---|---|
| MMB [3] | 54.7 | 57.1 | 59.5 | 61.9 | 64.9 | 67.8 | 71.1 | 72.6 | 75.9 |
| LINE [37] | 55.0 | 58.6 | 66.4 | 73.0 | 77.6 | 82.8 | 85.6 | 88.4 | 89.3 |
| Node2Vec [16] | 55.9 | 62.4 | 66.1 | 75.0 | 78.7 | 81.6 | 85.9 | 87.3 | 88.2 |
| DeepWalk [31] | 56.0 | 63.0 | 70.2 | 75.5 | 80.1 | 85.2 | 85.3 | 87.8 | 90.3 |
| TADW [49] | 86.6 | 88.2 | 90.2 | 90.8 | 90.0 | 93.0 | 91.0 | 93.4 | 92.7 |
| CENE [36] | 72.1 | 86.5 | 84.6 | 88.1 | 89.4 | 89.2 | 93.9 | 95.0 | 95.9 |
| CANE [40] | 86.8 | 91.5 | 92.2 | 93.9 | 94.6 | 94.9 | 95.6 | 96.6 | 97.7 |
| WANE [34] | 91.7 | 93.3 | 94.1 | 95.7 | 96.2 | 96.9 | 97.5 | 98.2 | 99.1 |
| Naive-VAE | 60.2 | 65.6 | 67.8 | 77.8 | 80.2 | 83.8 | 87.7 | 88.1 | 90.1 |
| VGAE [22] | 63.9 | 72.1 | 74.3 | 80.9 | 84.3 | 86.0 | 88.1 | 88.7 | 90.5 |
| PWA | 92.2±0.6 | 93.8±0.3 | 95.6±0.4 | 96.4±0.3 | 96.8±0.2 | 97.4±0.3 | 97.7±0.3 | 98.4±0.2 | 98.9±0.2 |
| VHE | **94.4±0.3** | **96.5±0.3** | **97.6±0.2** | **97.7±0.4** | **98.3±0.2** | **98.4±0.2** | **99.0±0.3** | **99.1±0.2** | **99.4±0.2** |

Table 6: AUC scores for link prediction on the HEPTH dataset.

| % Training Edges | 15% | 25% | 35% | 45% | 55% | 65% | 75% | 85% | 95% |
|---|---|---|---|---|---|---|---|---|---|
| MMB [3] | 54.6 | 57.9 | 57.3 | 61.6 | 66.2 | 68.4 | 73.6 | 76.0 | 80.3 |
| LINE [37] | 53.7 | 60.4 | 66.5 | 73.9 | 78.5 | 83.8 | 87.5 | 87.7 | 87.6 |
| Node2Vec [16] | 57.1 | 63.6 | 69.9 | 76.2 | 84.3 | 87.3 | 88.4 | 89.2 | 89.2 |
| DeepWalk [31] | 55.2 | 66.0 | 70.0 | 75.7 | 81.3 | 83.3 | 87.6 | 88.9 | 88.0 |
| TADW [49] | 87.0 | 89.5 | 91.8 | 90.8 | 91.1 | 92.6 | 93.5 | 91.9 | 91.7 |
| CENE [36] | 86.2 | 84.6 | 89.8 | 91.2 | 92.3 | 91.8 | 93.2 | 92.9 | 93.2 |
| CANE [40] | 90.0 | 91.2 | 92.0 | 93.0 | 94.2 | 94.6 | 95.4 | 95.7 | 96.3 |
| WANE [34] | 92.3 | 94.1 | 95.7 | 96.7 | 97.5 | 97.5 | 97.7 | 98.2 | 98.7 |
| Naive-VAE | 60.8 | 65.9 | 68.1 | 78.0 | 80.7 | 84.1 | 88.8 | 88.9 | 90.5 |
| VGAE [22] | 65.5 | 72.0 | 74.5 | 81.1 | 85.9 | 86.4 | 88.4 | 88.8 | 90.4 |
| PWA | 92.8±0.5 | 94.2±0.3 | 96.1±0.4 | 96.7±0.3 | 97.6±0.2 | 97.8±0.2 | 97.9±0.2 | 98.6±0.3 | 99.0±0.2 |
| VHE | **94.1±0.4** | **96.8±0.3** | **97.5±0.3** | **98.3±0.2** | **98.3±0.3** | **98.5±0.3** | **98.8±0.2** | **99.0±0.2** | **99.4±0.3** |

Table 7: AUC scores for link prediction on the ZHIHU dataset.

| % Training Edges | 15% | 25% | 35% | 45% | 55% | 65% | 75% | 85% | 95% |
|---|---|---|---|---|---|---|---|---|---|
| MMB [3] | 51.0 | 51.5 | 53.7 | 58.6 | 61.6 | 66.1 | 68.8 | 68.9 | 72.4 |
| LINE [37] | 52.3 | 55.9 | 59.9 | 60.9 | 64.3 | 66.0 | 67.7 | 69.3 | 71.1 |
| Node2Vec [16] | 54.2 | 57.1 | 57.3 | 58.3 | 58.7 | 62.5 | 66.2 | 67.6 | 68.5 |
| DeepWalk [31] | 56.6 | 58.1 | 60.1 | 60.0 | 61.8 | 61.9 | 63.3 | 63.7 | 67.8 |
| TADW [49] | 52.3 | 54.2 | 55.6 | 57.3 | 60.8 | 62.4 | 65.2 | 63.8 | 69.0 |
| CENE [36] | 56.2 | 57.4 | 60.3 | 63.0 | 66.3 | 66.0 | 70.2 | 69.8 | 73.8 |
| CANE [40] | 56.8 | 59.3 | 62.9 | 64.5 | 68.9 | 70.4 | 71.4 | 73.6 | 75.4 |
| WANE [34] | 58.7 | 63.5 | 68.3 | 71.9 | 74.9 | 77.0 | 79.7 | 80.0 | 82.6 |
| Naive-VAE | 56.5 | 59.0 | 60.2 | 60.9 | 62.5 | 66.4 | 68.1 | 68.3 | 69.0 |
| VGAE [22] | 55.9 | 59.1 | 61.9 | 62.3 | 64.6 | 67.2 | 70.1 | 70.6 | 71.2 |
| PWA | 62.6±0.3 | 67.5±0.2 | 70.8±0.1 | 72.2±0.2 | 77.1±0.2 | 80.3±0.2 | 80.8±0.2 | 81.9±0.2 | 83.3±0.1 |
| VHE | **66.8±0.4** | **71.3±0.3** | **74.1±0.2** | **75.2±0.1** | **81.6±0.2** | **84.1±0.2** | **84.7±0.2** | **85.8±0.2** | **86.4±0.2** |