[Reviews · NeurIPS 2019]

Reviewer 1



CLARITY: Overall the paper is clearly written. There were few parts that I had hard time understanding (i.e., such as line 223-224, how can q(z_i|x_i, x_j, w_ij) denotes the posterior with and without edge information? I guess it's when q_0, and when q_1?) The authors seemed to have squeezed the format a bit too much. There's a lot of details, might be better to focus on key contributions and make the paper self-containing. For example, Figure 2, as is, impossible to decipher and we have to look up the supplementary material to understand. Overall, I found it hard to tell apart the contribution of homophilic prior on the performance. The particular design choice of encoder and decoder architecture (section 3.2), as well as how to set the global network embeddings (section 3.3). A proper ablation study should be designed to test where the improvement is coming from. I assume maybe the PWA baseline that the author is proposing is the proposed method minus the homophilic prior? But it wasn't super clear. And then the delta between PWA and VHE is fairly small... The model introduces the notion of link, no link, and unknown, but then the experiment doesn't show whether modeling of unknown brings meaningful differences? ==== AFTER THE AUTHOR RESPONSE: Thanks for careful response and pointing out the ablations. It is much clearer with the author response. Though, about (d), the figure doesn't seem to support that modeling unknown link brings significant gain..

Reviewer 2



[I read the author response; my score remains unchanged] The writing of the paper is very clear, and the paper does a good job of motivating the method and describing the technical details of the approach. I did not find any issues with the technical derivations for the VAE formulation, and to my knowledge the proposed VHE method is novel. The proposed method seems very practical, especially because it can tolerate incomplete graphs, missing edges, and new nodes that appear at test time. The authors include their phrase-to-word alignment system among their list of contributions, but that portion of the work appears less novel. There have been a number of methods that rely on mutual attention between two sequences of tokens (see e.g. https://arxiv.org/pdf/1606.02245.pdf ; https://arxiv.org/pdf/1611.01603.pdf), and to the extent that the authors' implementation has incremental differences over the past work there is no indication that these differences affect accuracy. That said, the word representation method is not the key focus of the present paper. [Regarding the response: my initial reaction reading the paper for the first time was that the authors were claiming novelty of the architecture itself; I agree that the application of such an architecture to this specific task is separate and can be considered a contribution of this paper.] [Thank you for promising to address some of the presentation issues below] 77: Why is it "without loss of generality"? I don't follow. 190: Are K_r/K_c a function of L_j/L_i, or are they fixed hyperparameters? I think "number of filters" generally refers to the channel dimension, but these are sequence length dimensions instead. In general the text in 183-195 isn't as clear as the remainder of the paper. Grammar and formatting: 20: its -> their 265: What does PWA stand for? Table 2: Could you say more about the methods? For example, adding columns to indicate whether each method encodes topology, content, both, is a discriminative method, is a generative method. Defining the acronyms might also help. 295: "When VHE works?" -> "When does VHE work?" 344: grammar seems a bit odd at the phrase "are not necessary". Maybe "documents with similar info do not necessarily cite each other", or "documents citing each other do not necessarily have similar info" (depending on what the intended meaning is) [349: textural -> textual (?)]

Reviewer 3



This paper uses the formalism of autoencoders to model the joint probability distribution of the text attributes of a _ pair of nodes _. The correlation/dependence is encapsulated by the corresponding pair of latent variables, in the case where the two nodes are connected by an edge. In a first derivation (7), the edge is assumed visible. Later (8), it is assumed latent and marginalized. The manifest advantage of this approach is that the information about edges and the text attributes of the vertices are encoded simultaneosuly, unlike previous approaches as reviewed by the authors. The parameters of the posterior distributions are estimated using an original textual embedding which takes into account the text of the other nodes (Section 3.2). Even just this embedding alone leads to significant accuracy improvements for an existing SOTA method (CANE; the integrated method is noted as PWA in the tables). I only have a somehow major remark about the presented approach. The authors claim that this method is able to encode "structural information" per node. This means, create an embedding of its connectivity to other nodes. In reality, the embedding (h) used in the approach is just an arbitrary vector per node that is learned by minimizing the training objective. How could this be proved to be connectivity information? The authors also state that they are able to mollify the computational complexity deriving from modelling pairs of vertices. It would be very useful if they added actual computational times for both training and inference (they are missing also from the Supplementary). ==== AFTER THE AUTHOR RESPONSE: I am overall satisfied with the authors' response. About H, I had no doubt that it is useful (see Fig. 4(d)): whether it comes into correspondence with actual structural information is still to be proved.

[Author Response · NeurIPS 2019]

We thank all the reviewers for their insightful and constructive comments, and answer their questions below.

**To Reviewer #1**

(a) *line 223-224*: Yes, conditional posteriors are used here. We use $q_1$ if $w_{ij} = 1$ and $q_0$ otherwise.

(b) *Format squeezed too much, Fig. 2 difficult to decipher*: We will update Fig. 2 to improve clarity. We will also make edits to highlight the key contributions and move less relevant details to the Appendix.

(c) *What's PWA and clarify VHE's gain over PWA:* PWA refers to WANE with phrase-by-word alignment for textual feature extraction. It is a discriminative model (no prior of any sort) while our VHE is a generative solution. Generative baselines without homophilic priors are naive-VAE and VGAE. Tables 2 and 3 contain the ablation study requested by the reviewer, which decomposes the gains into individual contributions. PWA improves over WANE (prior SOTA), showing the proposed phrase-by-word alignment (a side contribution ) delivers better performance. VHE's gain over PWA is more apparent on vertices with fewer connections (see Fig. 3), which demonstrates VHE's robustness and effectiveness. This also bears practical significance because low-degree vertices are what existing models struggle with.

(d) *Whether modeling of unknown links brings meaningful differences in experiments:* This corresponds to the ablation study provided in line 332-336 and Fig. 4(c). We have a hyper-parameter $\alpha$ to control the strength of uncertainty and observe that a proper choice of $\alpha$ (0.4) achieves the best results.

(e) *Limitations and prospects*: While achieving significant performance gains, the current setup of VHE only encapsulates pairwise structural information in the prior. The integration of higher-order topological information is an interesting topic, and we leave it for future investigation.

**To Reviewer #2** We appreciate reviewer's acknowledgement of our novelty and constructive suggestions provided.

(a) *Contribution of phrase-to-word alignment:* While the key contribution of this work is the VHE model, our phrase-to-word alignment module also demonstrates significant performance gains over existing SOTA, which qualifies it as a side contribution. While several similar sequence-to-word attention mechanisms have been considered in other NLP tasks, the application in a network embedding context is novel.

(b) *What's "without loss of generality" in Line 77*: We mean the techniques developed can be similarly applied to directed graphs. We will clarify this in our revision.

(c) *Clarify Line 190, Line 183-195*: $K_r/K_c$ are fixed hyper-parameters shown in Line 525-526. We will revise Line 183-195 to improve clarity.

(d) *What's PWA*: PWA refers to WANE with the proposed Phrase-by-Word Alignment. See our reply (c) to Reivewer #1 for additional details.

(e) *Improving Table 2*: Thanks for the suggestions. We will categorize the methods in Table 2 into four groups, namely topology-only baselines, topology+content baselines, generative baselines and proposed models. Table 2 will be revised accordingly, and acronyms will be clearly defined.

(f) *Response to Improvements*: We agree that the current VHE implementation fails to capture higher-order information and does not account for global topology. Extensions to these directions are interesting topics, which we are actively exploring. To note, we experimentally found that for textual network applications a fully generative solution encodes too much nuisance information, which is often detrimental to the performance, thus pooling is applied.

(g) We will fix all the grammar and formatting issues pointed out by the reviewer.

**To Reviewer #3** We thank the reviewer for the positive reviews. The remarks raised are addressed below.

(a) *Why $\boldsymbol{H}$ encodes connectivity information?* The use of structural embedding $\boldsymbol{H}$ is motivated from Node2Vec [17], where it is assumed vertex-based topological profile (i.e., structure) can be encoded by a learnable vector representation. To verify $\boldsymbol{H}$ indeed captures structural information, we carried out an ablation study and summarized the results in Figure 4(d). It is clear that the use of structural embedding $\boldsymbol{H}$ improves over models that only use text information when predicting network topology. We will further clarify this.

Table 1: Computational cost for VHE.

| Dataset | Train (s/epoch) | Inference (s) |
|---------|-----------------|---------------|
| Cora | 2.8 | 45.6 |
| HepTh | 1.6 | 17.2 |
| Zhihu | 17.8 | 500 |

(b) *Actual computational cost.* The computational costs are charted in Table 1. This confirms VHE is very efficient in practice, and the significant performance gain fully justifies the mild increase in computation time comparing to existing SOTA. A more comprehensive discussion will be added to our revision.

(c) *Clarifications.* We will clarify that textual attributes are still available for missing vertices. (Line 227) We use 50 Monte Carlo samples to reduce the computational cost for global network embedding with each vertex.

[Meta-Review · NeurIPS 2019]

The reviewers all agreed that this was an at least acceptably strong paper. It is clearly written and has strong results. The author's response appears to address the issues raised by R3.